# GPU Rasterization-Based 3D LiDAR Simulation for Deep Learning

**DOI:** 10.3390/s23198130

**Published:** 2023-09-28

**Authors:** Leon Denis, Remco Royen, Quentin Bolsée, Nicolas Vercheval, Aleksandra Pižurica, Adrian Munteanu

**Affiliations:** 1Department of Electronics and Informatics (ETRO), Vrije Universiteit Brussel, Pleinlaan 2, 1050 Brussels, Belgium; remco.royen@vub.be (R.R.); qbolsee@etrovub.be (Q.B.); adrian.munteanu@vub.be (A.M.); 2Imec, Kapeldreef 75, 3001 Leuven, Belgium; 3Department of Telecommunications and Information Processing (TELIN-GAIM), Ghent University, 9000 Ghent, Belgium; nicolas.vercheval@ugent.be (N.V.); aleksandra.pizurica@ugent.be (A.P.); 4Department of Electronics and Information Systems, Clifford Research Group, Ghent University, 9000 Ghent, Belgium

**Keywords:** LiDAR, GPU, simulation, data generation, neural networks

## Abstract

High-quality data are of utmost importance for any deep-learning application. However, acquiring such data and their annotation is challenging. This paper presents a GPU-accelerated simulator that enables the generation of high-quality, perfectly labelled data for any Time-of-Flight sensor, including LiDAR. Our approach optimally exploits the 3D graphics pipeline of the GPU, significantly decreasing data generation time while preserving compatibility with all real-time rendering engines. The presented algorithms are generic and allow users to perfectly mimic the unique sampling pattern of any such sensor. To validate our simulator, two neural networks are trained for denoising and semantic segmentation. To bridge the gap between reality and simulation, a novel loss function is introduced that requires only a small set of partially annotated real data. It enables the learning of classes for which no labels are provided in the real data, hence dramatically reducing annotation efforts. With this work, we hope to provide means for alleviating the data acquisition problem that is pertinent to deep-learning applications.

## 1. Introduction

The field of AI has witnessed rapid development in recent years [1,2,3]. The increasing flexibility of GPUs and their ability to consume vast amounts of data have led to the birth of deep neural networks and have stirred the AI landscape. This is more specifically also the case for the domain of Autonomous Vehicles, where artificial-intelligence-based solutions are indispensable [4,5,6,7,8,9,10]. One can regard the development of deep neural networks (DNNs) as a two-phased process: designing the architecture and acquiring and annotating data. The majority of research focuses on the former. This can be explained by the high-quality datasets publicly available but also by the challenge imposed by acquiring and annotating data. Though publicly available datasets are suitable for obtaining generalized models, the industry often has specific requirements and prefers specialized solutions that work optimally with their hardware. This, in turn, imposes obtaining specific, sometimes non-existent, annotated datasets. This is especially true for LiDARs, as hardware specifications can differ greatly. It has been shown that not taking the specifications of the LiDAR into account during training results in decreased performance due to sensor bias [11]. The main question then becomes how to acquire and annotate LiDAR-specific data. One has the choice between annotating real-world samples or generating synthetic data. Both have their merits and flaws. When leveraging real data, no mismatch exists between the training and test sets. However, compiling sufficiently large real-world datasets is very challenging given the enormous amount of accurately labelled data required to train reliable models. When performing more advanced tasks, such as segmentation, manual annotation becomes borderline unfeasible, if not impractical. An overview of the processes involved for obtaining high-quality annotated datasets is provided in [12]. The processes rely on very precisely calibrated and synchronized hardware combined with state-of-the-art traditional parsing methods and deep-learning solutions. After processing, roughly 30% still needs to be labelled manually, which, in turn, almost guarantees the introduction of human errors. Lastly, real-world datasets tend to be prone to class imbalance. For instance, on average, cars are much more represented in traffic scenarios compared to other vehicles [13,14,15,16,17]. By contrast, simulated data do not suffer from any of the aforementioned limitations and can be used to generate well-balanced, perfectly annotated datasets. However, deviations from reality pose significant risks, and mismatches between training and test sets must be limited and closely observed.

This paper describes a novel GPU-accelerated Time-of-Flight (ToF) sensor simulator. The idea stems from the desire to apply deep-learning solutions for a custom LiDAR (Light Detected and Ranging) for which no annotated data are available. Though focusing on AI for solid-state LiDAR, we introduce concepts that are generic and can provide means to enable deep-learning on other sensing devices for which no annotated data are available. From a technical perspective, the proposed simulator harvests the computational power of the GPU by exploiting the real-time graphics rendering pipeline, hence significantly increasing data generating speeds compared to the state-of-the-art ToF simulators, which rely on ray casting [18,19]. Experiments reveal that the proposed simulator mimics reality accurately and is suitable for generating datasets to train neural networks. Summarised, our main contributions are as follows:The proposed simulator exploits the highly optimised rasterization pipeline of the GPU, increasing data generation speeds one hundredfold compared to the state-of-the-art. This is particularly useful for data-driven applications demanding vast amounts of data, such as deep-learning.The presented algorithms are generic and allow simulation of any ToF sensor, including those with unique sampling patterns, which has not been considered in any prior work to the best of our knowledge.We introduce a novel loss function leveraging synthetic and partially annotated real data to alleviate the mismatch between simulation and reality. It furthermore allows the model to learn classes for which no labels are provided in the real-world training set.Our techniques greatly alleviate, and in some cases completely eliminate, the tedious annotation process of real data, even for difficult tasks such as segmentation, unlike any prior work focusing on LiDAR. That is, we present a working pipeline for training reliable models for specific ToF devices with unique hardware specifications when no or limited annotated data are available.Two neural networks operating on real data for denoising and semantic segmentation are trained using synthetic point clouds generated by a digital twin of a real-world prototype LiDAR.

The remainder of this paper is organized as follows. Related works are presented first. Next, the proposed methodology and implementation details of the digital twin are detailed. This is followed by the experimental evaluation which discusses two deep-learning applications. Limitations are discussed in Section 5, which is followed by the conclusions of our work.

## 2. Related Work

AI for LiDAR has gained a lot of interest in recent years. The active developments in the research domain are most likely fuelled by the high interest expressed by the autonomous navigation industry. Most of those developments employ real-world data captured using a physical LiDAR [20,21,22,23,24,25,26,27,28]. However, realistic and performant LiDAR simulations are currently highly desirable given the difficulties with data gathering and annotation. Much effort has therefore been made to create accurate ToF simulations. One can classify them as either being ray casting, rasterization, or AI-based.

**Ray-casting-based** techniques naturally stem from the fundamental principles of LiDARs, where multiple laser pulses are emitted to capture depth measurements. These simulators leverage ray casting algorithms, which have a well-established history in the field of computer graphics and were originally employed for tracing light rays to generate realistic imagery [29]. Given their intrinsic connection with the underlying hardware, it is unsurprising that the majority of LiDAR simulators fall within this category. Early works primarily concentrated on airborne LiDARs for terrestrial applications. Initial efforts predominantly focused on characterizing the properties of reflected laser beams, which is a crucial factor in determining the position of leaves during canopy scanning [30]. Subsequently, more attention was directed towards considering atmospheric effects on the laser beams [31]. With the advent of increased computational power, higher-resolution 3D models were incorporated, yielding more precise simulations for both terrestrial and urban scanning [32,33]. Later on, the fidelity of the underlying physical simulations improved through the integration of radiative transfer models into the simulations [34]. As these simulations became more accurate, virtual simulations were employed to determine the optimal scanning paths, particularly for canopy scanning to maximize leaf area coverage [35]. Later on, GPU flexibility increased and significant acceleration of ToF simulations was achieved through hardware-accelerated ray casting [36].

The recent emergence of autonomous driving has significantly redirected research efforts towards the automotive domain, particularly with a strong emphasis on the integration of deep-learning techniques. In the study presented in [37], the authors propose a ray tracing simulation within a traffic environment as a means to develop occupancy grid mapping algorithms. Their simulation’s validity is substantiated through a virtual reconstruction of a real-world environment. Building upon this concept, ref. [38] takes a step further by employing video games in conjunction with ray-casting-based LiDAR simulations to generate simulated traffic data to train neural networks. A similar philosophy is embraced in [19], where the focus lies on constructing a traffic simulator using the Unreal Engine [39]. The authors’ platform supports various sensors, including ray-casting-based LiDAR [39]. Subsequently, this simulator is utilized for training neural networks geared towards autonomous driving applications [40]. The use of synthetic data was also adopted by [41], in which real-world data are employed to enhance the accuracy of the physical properties of virtual environments and actors, such as reflected beam intensities and noise.

The rising success of data-driven AI for point clouds in turn led to the development of publicly available ray-casting-based ToF simulators. One of the pioneering efforts in the field was Helios [42]: an all-purpose ToF sensor simulator implemented in Java. Subsequently, an enhanced version was developed in C++, which led to a substantial reduction in simulation time, as discussed in [43]. More recently, the latest generic ToF simulator to emerge is Blainder [18], a simulator integrated with Blender [44]. Blainder offers the capability to generate meticulously annotated synthetic point data, marking a significant advancement in ToF simulation technology.

The second category of virtual ToF sensors comprises **rasterization-based** methods. Such simulators harvest the power of the GPU by exploiting the 3D rendering pipeline popularized by real-time graphics applications. The principle idea is to parallelize the computation of the first hit of the laser pulses by reverse-transforming the values of the depth map. The main advantage over the aforementioned ray-casting-based techniques is the dramatic reduction in computation time due to massive parallelization.

One of the earlier rasterization-based LiDAR simulators is discussed in [45]: it introduces a novel approach involving blending real-world backgrounds with synthetic actors, which are subsequently synthesized into the final images. This simulation method entails simulating a virtual LiDAR by initially projecting the virtual world onto cube-maps, which are then sampled. A similar methodology is employed in [46], where a combination of real-world data and synthetic actors is achieved through surface splatting techniques. In contrast, the work presented in [47] exclusively relies on virtual data for LiDAR simulation. This approach employs equirectangular rendering and utilizes spherical projection and wrapping techniques to expedite the simulation process. Like the previously mentioned methods, it necessitates rendering the entire virtual world prior to simulation. Lastly, the research detailed in [48] introduces a LiDAR simulation technique that involves the insertion of virtual actors into real point clouds. This approach updates the existing rays within the original data by identifying ray correspondences through laser projection onto the synthetic image, offering a unique perspective on LiDAR simulation.

This work also belongs to this group, but it differs substantially from the aforementioned methods in the sense that it generalizes the ToF simulation. First, it allows for the virtually mimicking of the behaviour of both solid-state and electromechanical LiDARs. By contrast, the aforementioned citations only consider the latter and are not suitable for solid-state devices. Secondly, the unique sampling pattern is not accounted for in any existing works. In other words, no prior work enables the implementation of an exact digital twin, which is crucial for doing realistic simulations. Furthermore, the proposed simulator does not require the whole virtual world to be rendered for the simulation. Only portions visible by the LiDAR are rendered, thus significantly increasing simulation efficiency. Though those works can indeed be improved upon, it is fair to mention that they do not focus on the simulation. It is simply a byproduct for data augmentation. Simply put, the proposed methodology has the flexibility, realism, and generality of ray-casting-based simulators with the additional benefit of the computational efficiency imposed by the rasterization-based solutions.

Most recent advancements in the field have endeavoured to harness deep-learning techniques for **AI-based** simulating of LiDAR. One noteworthy contribution is exemplified by UniSim [49]. In this research, deep neural networks are applied to generate novel training data, accommodating dynamic actor modifications and limited sensor adjustments. A comparable approach is adopted in [50], which primarily concentrates on introducing new dynamic actors into real-world-captured footage. In contrast, ref. [51] takes a distinctive approach by enhancing the attributes of synthetic ray-casting-based LiDAR simulation through neural networks. This enhancement enables a more accurate representation of physical characteristics such as noise, reflected intensity, and dropped points. However, it is imperative to note that all these methodologies necessitate high-quality real-world data for training the neural networks, despite their ability to yield highly realistic results.

All related works have in common that they alleviate the **data acquisition problem**. Recent studies confirm that acquiring and labelling data indeed poses one of the main challenges for reaching Level 5 vehicle autonomy [52,53]. This is particularly more challenging for LiDAR, as each device has it own sampling pattern, noise model, and pulse intensity measurements. Ignoring such parameters is possible, and potentially mismatched training data can be extracted from existing datasets [12,54,55]. However, another solution is necessary when the required data simply does not exist. Prior to ApolloScape [12], no labels for flora were available for point clouds. The increasing realism of real-time graphics and the required efforts of data gathering have motivated researchers to leverage virtual simulators for both unmanned aerial vehicles [56,57] and autonomous driving [58,59]. Recent works in the field of deep-learning have also benefited from such simulators [60,61,62,63]. Others synthesize training samples by combining real data with virtual actors [45,64]. To the best of our knowledge, all previous works involving LiDAR simulations require real annotated data for producing acceptable performance [38,40,41,46,60,61,62]. We will show later that this is also true for one of our applications. However, unlike any previous work, we will introduce a novel training methodology in which labels for new classes can be completely omitted in the real training data, thus significantly reducing annotation efforts.

## 3. GPU-Accelerated LiDAR Simulation

This section thoroughly discusses the proposed GPU-accelerated simulator. We start by briefly explaining the underlying mechanics of ToF sensors and by providing generic algorithms suitable for simulating such sensors. Thereafter, we explicate the implementation of a virtual digital twin based on the hardware specifications of its real world counterpart.

### 3.1. Basic Principles ToF Sensors

Figure 1 illustrates the fundamental principles of a ToF sensor. At its core, a ToF sensor comprises a cost-effective Complementary Metal-Oxide-Semiconductor (CMOS) pixel array sensor paired with an active infrared light source. Depth measurements are derived by illuminating the scene with modulated light and measuring the time it takes for the reflected light to return to the CMOS sensor. These time measurements are subsequently translated into depth values. The modulated light source typically encompasses projected infrared laser beams, with each laser pulse yielding a depth measurement. ToF cameras, such as the Microsoft Kinect v2, construct depth maps wherein depth measurements are organized in a rectangular grid format, analogous to an image. Conversely, LiDARs often employ distinctive sampling patterns, resulting in the creation of point clouds.

As mentioned before, the projection of rays originating from a light source has been extensively studied in the past, particularly in the context of computer-generated imagery and notably within the realm of ray tracing. Considerable effort has been dedicated to optimizing ray casting algorithms due to the substantial time investment required for generating realistic images, which involves tracing millions of light rays [65]. Simulation of ToF sensors can be achieved by adapting these methods, effectively simulating the firing of rays into the virtual scene for each laser pulse to retrieve depth information. Given the close correspondence between the simulation process and the physical ToF device, it is unsurprising that a significant portion of ToF simulators rely on such ray casting methodologies. As far as our knowledge extends, all generic ToF simulators adopt a ray-casting-based approach, including examples like Helios [42], its improved version [43], and Blainder [18]. These simulators are strictly implemented on the CPU and often cast rays either sequentially or with limited parallelization.

### 3.2. Rasterization-Based ToF Sensor Simulation

Unlike ray-casting-based methods, which sequentially simulate laser pulses, the proposed method simulates *all* rays in parallel by exploiting the rasterization pipeline of the GPU for determining the first-hit of each laser pulse. More specifically, our simulator reverses the forward 3D rendering pipeline by employing data stored in the framebuffer for reconstructing the input geometry of the virtual 3D space in a point-wise manner, hence simulating the behaviour of ToF sensors. A simplified version of the pipeline is depicted in Figure 2. The blue blocks in the figure indicate reprogrammable stages of the rendering pipeline. As will be explained later, the proposed simulator only requires the use of the vertex and fragment shaders, making it compatible with older hardware. In a general context, the task of the vertex shader is to project vertices from local coordinate space to the image plane using a series of affine transformations. The transformed vertices are passed to the geometry shader, after which the input polygons are rasterized, colourized, and clipped in order to fit the image plane. The geometrical transformations applied to each input vertex in the forward rendering pipeline can be mathematically expressed as
(1)$$p_{i}=\upsilon \left(M_{proj}\cdot M_{view}\cdot M_{model}\cdot p_{l}\right),$$
with $p_{l}$ representing an input vertex in its local coordinate system, also referred to as model space. $M_{proj}$, $M_{view}$, and $M_{model}$ denote the projection, view, and model matrices, respectively. The variable υ transforms data from clipping space to image space. The resulting point $p_{i}$ is a homogeneous coordinate on the image plane, with *z* corresponding to the z-buffer value. The proposed simulator computes the first hit of cast rays by partially reversing the chain of transformations to obtain points in camera space: (2)$$p_{c}=M_{proj}^{-1}\cdot \upsilon^{-1}\left(p_{i}\right).$$

As will be explicated later in Section 3.3, the projection matrix must be constructed following specific hardware guidelines dictated by the simulated ToF sensor and must remain constant through the simulation. We therefore compute $M_{proj}^{-1}$ once on the CPU and make it available to the fragment shader rather than computing it on the GPU.

Synthetic point clouds are produced through the process of rendering a specific three-dimensional scene via the forward rendering pipeline, yielding a 32-bit floating-point texture as the output. In accordance with standard practices, the primary responsibility of the vertex shader lies in the projection of input vertices from the virtual world onto the image plane, as dictated by Equation (Equation 1). Subsequently, the simulation of LiDAR rays is carried out at pixel-level granularity within the fragment shader. This simulation involves the partial reversal of the transformation sequence. Specifically, camera space coordinates are obtained using the corresponding pixel’s image space coordinates together with the inverse projection matrix and reversing the forward rendering pass by applying the reverse transformations according to Equation (Equation 2). Depth information can be obtained directly since the coordinate system’s origin and the camera’s position coincide. Throughout this procedure, all other attributes pertaining to the point cloud are calculated at pixel-level in the fragment shader and are subsequently stored in the framebuffer object. After rendering, the output texture is retrieved from the GPU to the main memory, from which point clouds are constructed by associating on a per-pixel basis the camera space coordinates with their respective point attributes. Every constructed point corresponds to a laser pulse of the virtual LiDAR, with its position coinciding with that of the camera. The camera thus acts as the ToF sensor.

### 3.3. Matching Hardware Specifications

To realistically mimic real hardware, camera parameters should be selected according to sensor specifications. The aspect ratio, rendering resolution, and Field-Of-View (FOV) must perfectly match that of the sensor. Near and far planes must br chosen to reflect the minimum and maximum sensing range, respectively. The following perspective-projection matrix can serve that purpose: (3)$$M_{proj}=\left[\begin{array}{cccc} \frac{h}{w\tan(\alpha /2)} & 0 & 0 & 0\\ 0 & \frac{1}{\tan(\alpha /2)} & 0 & 0\\ 0 & 0 & \frac{-(f+n)}{f-n} & \frac{-2fn}{f-n}\\ 0 & 0 & -1 & 0 \end{array}\right].$$

This is derived from the standard projection matrix commonly used in real-time hardware-accelerated rendering. The variables *w* and *h* symbolise the width and height of the framebuffer, respectively. The near and far planes are denoted by *n* and *f*, respectively. The vertical FOV is represented by α. $M_{proj}$ projects the virtual input geometry on the image plane according to the exact specifications of the simulated ToF sensor. As elaborated in Section 3, the output texture is subject to sampling after its retrieval from the GPU, which is a step integral to the point cloud construction process following forward rendering. To match hardware specifications, the image plane must be sampled akin to the depth-measuring pattern of the real-world device. The Microsoft Kinect v2, for example, produces depth maps of resolution 512×424. In this case, the output texture can match that resolution, after which pixel-wise sampling can be employed for point-data generation. LiDAR laser pulses typically are not as structured. To imitate the pattern of their laser projections, nearest-neighbour sampling can be done using uv-coordinates derived from hardware specifications. If such information is unavailable, one can point the LiDAR towards a flat surface and derive the uv-coordinates by normalizing the *x* and *y* values of the generated point cloud along their respective dimensions. Optionally, the utilization of higher-resolution rendering can be considered as a means to ensure that each laser pulse corresponds to a different pixel. However, it remains imperative to maintain the perspective projection as described by Equation (Equation 3). In our specific case, we render the output image at a resolution twice that of the real-world device. Figure 3 compares the results obtained through pixel-wise and uv-sampling. The image on the right mimics our LiDAR, for which uv-coordinates were derived from hardware specifications.

### 3.4. Noise Model

Real-world depth measurements of ToF devices are subject to noise. The measurement uncertainty is caused by numerous factors such as photon shot, dark current noise, readout noise, etc. [66]. Previous research has established that the perturbations to the depth values are normally distributed [66,67] and are dependent on the distance of the hit surface, its reflectance, and the ambient light. Though the exact noise model varies among sensors, all manifest noise along their respective rays. This characteristic is an inherent outcome of the underlying mechanics of ToF sensors. As the directions of the laser pulses are known, the perturbation $\vec{p_{n,i}}$ of a given point $p_{i}$ can be formulated as
(4)$$\vec{p_{n,i}}=p_{i}+N\left(0,Z\left(d_{i},\varphi_{i},l_{i}\right)\right)\vec{R_{e,i}},$$
with $\vec{R_{e,i}}$ denoting the vector from the origin of the emitter to $p_{i}$. $N(0,Z\left(d_{i},\varphi_{i},l_{i}\right))$ represents a random variable distributed normally with zero mean and variance $Z(d_{i},\varphi_{i},l_{i})$. The variables $d_{i}$, $\varphi_{i}$, and $l_{i}$ denote the depth, albedo, and ambient light, respectively, associated with $p_{i}$. An example obtained when applying noise on a point cloud generated by our digital twin is shown in Figure 4b. Clean point clouds such as the one depicted in Figure 4a may serve as ground truth for training neural networks for denoising purposes. Interestingly, they can never be acquired with real-world LiDARs—they can only be obtained through simulation using a virtual digital twin.

### 3.5. GPU Implementation Details

The close relationship of the proposed methodology with 3D hardware-accelerated rendering makes it highly suitable for a GPU implementation. In fact, our simulator can be almost fully processed by the programmable graphics pipeline. The only exception is the simulation of the unique LiDAR sampling patterns, which requires non-uniform reading of the output texture, which, in turn, dictates CPU processing. Nevertheless, in all cases, the computation of the first-hit point for each ray is carried out on the GPU, making use of its rasterization hardware. This approach yields a substantial reduction in execution time, as this particular step demands the most significant computational resources. We will further substantiate this assertion through our experimental evaluation, as outlined in Section 4. We emphasize that the reduction in execution time is particularly important for data-driven applications, such as deep-learning, which require vast amounts of data to produce reliable results.

As mentioned before, our implementation uses a GPU program that outputs for each pixel fragment its corresponding camera space position to a texture, which is subsequently retrieved to the main memory and is sampled to match the hardware specifications. In our implementation, z-buffer values are directly retrieved in the fragment shader as supported by the OpenGL Shading Language (GLSL) [68]. Not all GPU programming languages support this feature. If not supported, the depth texture must be generated using a custom GPU program in an additional rendering phase and be made available to the GPU program.

To avoid value clipping and quantization artefacts, 32-bit floating-point textures are employed. Using textures with limited bit-depth negatively impacts the quality of the produced point clouds. This is also true for the bit-depth of the z-buffer, which must be increased from the standard 16 bits to 32 bits. We note that this is supported by all current real-time-rendering APIs. Failure to use sufficient bit resolution in any point of the rendering pipeline results in aliasing artefacts, as demonstrated by Figure 5a. In our experiments, we noticed that a 32-bit process suffices to generate clean point clouds, such as the one shown in Figure 5b.

## 4. Experimental Evaluation

This section evaluates the performance of our simulator. We compare data generation speeds with the state-of-the-art and assess the validity of the synthetic data for two deep-learning applications, namely, denoising and semantic segmentation. We note that the main purpose of the discussed applications is to validate the simulator and to prove that training can be done successfully using the generated data.

### 4.1. Data Generation Speed

We compare our method with two different techniques from the literature: Blainder [18], which is the most recent ToF simulator for multiple sensors, and Carla [19], which is the state-of-the-art in driving simulation and also supports numerous sensors, including LiDAR. Execution times for different amounts of virtual laser pulses are summarized in Table 1. Experiments were carried out on a laptop with an Intel i7-8950 and an RTX 2070 max-q. The table shows that the proposed methodology allows generating point clouds consisting of 1 million samples at roughly 90 Hz. In other words, real-time high-resolution simulation can be achieved with relatively modest hardware. Resolution and execution times are linearly correlated for all methods except at very low resolutions for which rendering overhead may become dominant. Comparing with Blainder [18], we notice the average data generation time is reduced by a factor of 300. For Carla [19], the speed improvements add up to a factor of 100. The large performance increase of the proposed method is due to fully employing the GPU rasterization pipeline. In contrast, both Blainder [18] and Carla [19] rely on ray-casting methods. We do acknowledge that a GPU implementation of Blainder [18] or Carla [19] will narrow the gap in terms of execution times. We have therefore implemented a rudimentary simulator using NVIDIA Optix [69], which employs GPU-accelerated ray casting. Our simulator vastly outperforms this method as well, especially for low resolutions, where the overhead of building acceleration structures used by the ray-casting methods becomes dominant.

We emphasize that the results of the proposed simulation are identical to those produced by ray-casting techniques. That is, all methods presented in this section generate identical point clouds given the same sensor configurations and virtual environment. This is due to all of them being physically accurate. However, it is important to note that Carla [19] does not provide a general solution for ToF sensor simulation and that a solid-state LiDAR, for instance, cannot be simulated, unlike the proposed work. Therefore, both Blainder [18] and Carla [19] would benefit greatly by adopting the proposed method in terms of execution times. For Carla [19], this also comes with increased flexibility, allowing it to simulate any ToF sensor. Both findings demonstrate the value of the proposed techniques. Furthermore, our method is compatible with even the most basic GPUs, as no specialized hardware is required. Lastly, we remark that the proposed methodology can be integrated into any rasterization-based rendering engine, hence providing access to its modelling tools.

### 4.2. Point Cloud Denoising

The first deep-learning application considers spatial denoising. We discuss the data generation process and present results for real-world data.

#### 4.2.1. Dataset Generation

Training data are generated using the previously explained methodology. Ground truth is obtained by simply omitting the virtually added noise during point cloud generation. It is important to note that in this context, obtaining ground-truth data from the real-world device is not feasible due to the inherent noise always present in such devices, as mentioned previously. Generic meshes with varying geometrical features are employed to obtain robustness against all kinds of surface shapes. Point clouds are obtained through random scene compositions and by recovering the point data using our digital twin. For each point, KNN patches are extracted and an affine transformation is applied to locate the central point at the origin, with the laser pulse being aligned with the *Z*-axis of the coordinate system. Rotation is ignored, as it is not relevant for local denoising. Next, data are normalized independently along all axes to fit the [0, 1] interval. The normalized patches serve as input for the neural network, which outputs the offset along the *z*-axis with respect to the ground truth.

#### 4.2.2. Performance Evaluation

We evaluate the performance with real data through controlled calibration environments and a captured traffic scene. The first environment consists of the LiDAR pointing perpendicularly towards a wall and placed at a distance of 20 m. PointNet [70] is employed to perform the denoising task. The network has been trained using the training data previously mentioned and thus solely relies on synthetic data. Figure 6 illustrates the obtained point clouds before and after denoising. The figure clearly demonstrates the denoising capability of the network, which significantly improves the depth measurements of the LiDAR. Concretely, the root-mean-square error is reduced from 284.16 mm to 54.44 mm. The variance is reduced from 150.6 to 65.9 mm.

A second experiment positions three boards with known reflective properties at 100 m distance in an outdoor environment. An image of the setup is shown in Figure 7. Table 2 provides the standard deviation before and after the denoising. From the table, the denoising capabilities of the network are clear, which in turn proves the validity of the generated synthetic data and thus the simulator. As summarized by the table, noise is reduced by approximately 20%.

The third experiment comprises a real driving scenario. We only provide qualitative results as no ground truth data are known. Figure 8 shows a bird’s-eye view of two frames captured while driving. The colours indicate the intensity of the reflected laser pulses. After denoising, more structure is obtained in the point clouds. Edges are more refined, and driveable areas are more easily distinguishable. The vehicle on the left in Figure 8d is also more clearly defined compared to Figure 8c. From these observation, it is clear that the synthetic data generated by our digital twin can be leveraged to train neural networks for denoising purposes, thus, in turn, validating the proposed simulator.

### 4.3. Semantic Segmentation

To further assess our synthetic data, semantic segmentation is performed. Point clouds and annotations are extracted using a digital twin implemented in Unity [71].

#### 4.3.1. Dataset Generation

A rudimentary virtual world has been built to resemble the test data captured by our real-world LiDAR during a driving scenario. Figure 9 shows the 3D scene in more detail. Training data consist of 10,000 generated samples with cars randomly—but realistically—placed on the road. Camera parameters are slightly randomized to prevent overfitting and to obtain network robustness. Specifically, we enhance the network by introducing uniform noise within the range of −1° to 1° for the camera orientation. We apply the same principle to the spatial position, where uniform noise ranging from −0.1 to 0.1 m is introduced. This deliberate noise addition aims to prepare the network for handling variations in LiDAR placement that can occur in real-world settings. Projection parameters are retained and perfectly match those of the real LiDAR. Three different classes are used in our experiments: car, street, and flora. Point clouds are automatically annotated by rendering the segmentation map using colours corresponding to each class. We note that anti-aliasing features should be disabled for acquiring clean labels void of pixel blending. Figure 10 illustrates the different data layers associated with our annotated point cloud extraction method.

As mentioned previously, our simulator closely simulates the behaviour of the real-world LiDAR. Figure 11b shows a render mimicking the CMOS sensor of our LiDAR. The reference image in Figure 11a illustrates the resemblance with the synthetic data, both in terms of viewing parameters and in captured luminance. We note that the images are included for illustration purposes and that only depth information is used for point cloud generation. The choice is deliberate in order to void the incoherence of night and day when performing segmentation.

Unlike the previous application, we have enriched the training dataset with real-world *partially* labelled data. Annotations *for cars only* are obtained by inferring the CMOS image with a retrained Xception 65 model [72] trained on the Cityscape dataset [14] and transferring the labels from the CMOS image to the point clouds. Labels are manually corrected. The dataset enrichment is done to bridge the gap between reality and simulation. The incoherence in both datasets is not necessarily caused by the digital twin, but rather, it is due to the somewhat simplistic 3D scene, which strays from reality. Concretely, the training data comprise 10,000 artificial samples annotated with the classes car, street, and flora, and a small set of real data with only the cars labelled.

#### 4.3.2. Custom Loss

The annotation mismatch between real and synthetic data imposes a new training methodology: we propose a novel loss function that combines samples with differently labelled classes: (5)$$L=\left|\right|Y_{synth}-h\left(X_{synth}\right){\left|\right|}_{2}+\lambda ||Y_{real}-h\left(X_{real}\right){\left|\right|}_{2},$$
with $X_{real}$ and $X_{synth}\in R^{N\times 3}$ denoting the real and synthetic input point clouds, respectively. $Y_{synth}\in R^{N\times 3}$ symbolises the perfect synthetic semantic labels, whereas $Y_{real}\in R^{N\times 1}$ represents the real semantic car labels. The regularization factor is denoted by λ and the model by *h*. Since two prediction steps are required per optimisation step, each batch is subdivided into two sub-batches containing either real or synthetic samples. Predictions for each are retrieved thereafter. To guarantee processing the same number of synthetic and real samples, circular linked lists are employed. For each list, samples are reshuffled after processing the number of different samples in the list. The optimization step is done jointly to preserve smooth convergence.

#### 4.3.3. Performance Evaluation

We once more employ PointNet to test our approach. Purely for evaluation purposes, we obtain a test set containing 824 real-world samples annotated similarly to the real training set but with the inclusion of all categories instead of solely the cars. In this case, labels are not manually corrected, as this proved impractical. A representative sample is illustrated in Figure 12. Table 3 summarises the mean Intersection-over-Union (mIoU), overall accuracy (oAcc), and mean accuracy (mAcc) for three different experiments. The table indicates that solely relying on synthetic data leads to acceptable performance for flora and street. The IoU for cars shows room for improvement despite the high accuracy. Utilising only real data improves the IoU from 32.7 to 50.8. However, in this case, street and flora cannot be segmented, as such labels were not available during training. Employing the enriched dataset of 800 real samples dramatically improves the model, resulting in increased performance for all classes—even for street and flora, for which no extra labels are provided.

Visual results are presented in Figure 13. The CMOS image is included for reference. Results are obtained using the enriched dataset comprised of 1200 real-world samples. From the figure, it is clear that the trained network is able to segment the real captured point clouds well. Multiple cars are correctly and simultaneously segmented, as illustrated in Figure 13a,b. Minor confusion for the traffic sign is noticeable, which can be explained by such data not being present in the training set. Figure 13f shows a zoomed-in portion of the point cloud of Figure 13e. It illustrates the network being able to distinguish between car and road even when having a limited set of points. Overall, the street is accurately segmented as well. Most notable, the exit lane and T-junction in Figure 13c,d are accurately predicted even though such scenarios were not present in the virtual scene. Lastly, we also notice good inference for flora. Though perhaps not necessary in practice, similar as for the street, its prediction was freely obtained, as no manual annotation was involved. This example clearly shows the benefit of the proposed techniques.

As a last experiment, we provide results of an ablation study demonstrating the effect of the number of real-world samples with respect to prediction accuracy. Figure 14 summarizes the results. It is noteworthy that only 25 of such samples suffice for dramatically improving prediction accuracy, increasing the mIoU from 58.35 to 68.62.

The results presented here once more show the successful training of a deep network, which, in turn, validates the proposed simulator and proves that the generated data resemble reality close enough even for segmentation tasks, which are particularly sensitive to data mismatch.

## 5. Limitations and Discussion

Though the presented methodology very accurately mimics ToF sensors, a small deviation from reality exists. Similar to all the cited studies, be they ray-casting- or rasterization-based, we assume co-location of the sensor emitter and receiver, whereas in reality a baseline between both exists (2 cm in our case). This, in essence, ever so slightly perturbs the uv-coordinates of the laser pulses depending on the measured depths. It is also the reason why we advocate pointing the laser scanner perpendicular towards a vertical flat surface to extract the sampling pattern. Newly occluded points are also possible. A potential solution projects the resulting point cloud to the image plane of the receiver. However, this incurs a cost of a decrease in computational efficiency by a factor two with arguably no notable improvement in terms of network performance considering the very minor perturbations.

Another drawback arises from the utilization of synthetic data. We acknowledge that the preference should generally lean towards employing perfectly annotated real-world data, as bridging the gap between virtual and real data can be challenging. However, this approach is not always feasible, particularly when dealing with complex tasks like segmentation. One of the primary advantages of the proposed techniques is their ability to facilitate deep learning for ToF sensors when no real-world data are available. Nevertheless, as our experiments demonstrate, the inclusion of real-world data can enhance model accuracy. Therefore, a positive feedback loop can be established wherein initial predictions are recursively used as training data to further refine the model.

An additional strength of the proposed methods lies in their generality. They not only enable the creation of digital twins for virtually any ToF sensor but can also find applicability across various domains. While applications like autonomous driving may have relatively abundant data, other fields utilizing LiDAR, such as tree canopy estimation [73], crop classification [74], and wind farm efficiency prediction [75], may not possess readily available extensive datasets. The proposed methods could assist in simplifying the data collection and annotation processes in such contexts. Furthermore, even in domains with abundant data, our methods offer future-proofing benefits. Should the standard for traffic lights change, for example, new training data can be readily generated by updating the 3D model and running the simulator. This underscores one of the key advantages of employing synthetic data.

Lastly, while all the simulators discussed in this work generate the same point clouds under identical conditions, the reduction in data generation time should not be underestimated. This aspect holds significant importance, especially in data-driven applications like deep learning. The determination of class distributions in training data often necessitates multiple iterations to produce accurate models. Accelerating data generation by a factor of 100 can drastically reduce the resources required for obtaining such models, thereby saving time and money and reducing power consumption.

## 6. Conclusions

This paper presents a novel GPU-accelerated Time-of-Flight simulator. The proposed techniques are generic and can simulate any ToF sensor. Unlike any prior work, the proposed simulator allows the mimicking of the unique sampling patterns of each sensor. This enables the implementation of an exact digital twin, which is crucial for realistic simulations. Compared to the state-of-the-art, the proposed methodology decreases data generation times one hundredfold. It vastly outperforms GPU-accelerated ray-casting simulators as well. This is particularly important for data-driven applications, such as deep-learning, which often rely on enormous amounts of data. Using the presented methodology, a digital twin of a custom in-house LiDAR is devised and used for generating perfectly annotated datasets. Two applications are considered. A denoising neural network is trained that leverages only synthetic data and which reduces the root-mean-square error of the depth measurements for our wall calibration test from 284.16 mm to 54.44 mm. A second experiment shows a 20% reduction in variance for materials with different reflectivity. In terms of qualitative results, point clouds captured during a real-world driving scenario exhibit much more refined edges after denoising, making it easier to distinguish vehicles and drivable areas.

We further asses the proposed methods by training a deep network for semantic segmentation. Leveraging only synthetic data yields decent performance for two of the three classes. Dataset enrichment with real data is proposed to void the gap between reality and simulation. To enable training with the differently annotated samples, a novel loss function is introduced. It allows learning classes for which no labels are provided in the real training set, hence significantly reducing annotation efforts. An ablation study reveals that solely 25 *partially* annotated real samples suffice to already significantly increase prediction accuracy for *all* classes. When increasing the number of real-world samples with annotated cars to 800, our network is able to achieve an overall accuracy, mean accuracy, and mean intersection over union of 89.36%, 86.51%, and 75.64%, respectively when averaged over the classes car, street, and flora.

Both applications demonstrate the validity of our simulation and prove that the generated synthetic data resemble reality even close enough for training neural networks, which are particularly sensitive to data mismatch. However, despite addressing denoising and semantic segmentation, the proposed solutions are not limited to the discussed application domains. The simulator can be employed for research targeting LiDAR-based SLAM or vehicle navigation for example, whereas the proposed loss function can bridge the gap between synthetic and real data other than point clouds. Perhaps the proposed methods can find applications in thermal imaging, where not much data are available. Given the broad range of applications, we are convinced that the proposed techniques can help to alleviate the data acquisition problem, which is pertinent in many domains.

## Figures and Tables

**Figure 1 sensors-23-08130-f001:**
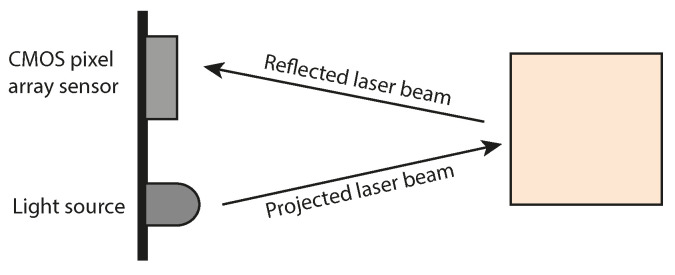
A visual schematic of a Time-of-Flight sensor. Depth measurement is achieved by emitting a laser beam into the scene and subsequently calculating the time required for the laser’s reflection to reach a complementary CMOS pixel array sensor.

**Figure 2 sensors-23-08130-f002:**
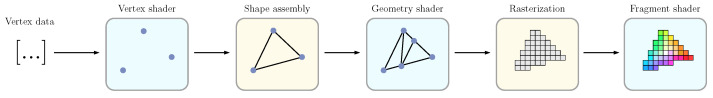
The 3D real-time rendering pipeline. Vertices are transformed to world space by the vertex shader, and faces are constructed. The embedded pixels defined through rasterization are coloured by the fragment shader. The proposed method constructs point clouds by reversing the pipeline. More specifically, the coordinates of each fragment are transformed back to camera space and stored in the fragment data, which is subsequently written to a texture and retrieved from the GPU.

**Figure 3 sensors-23-08130-f003:**
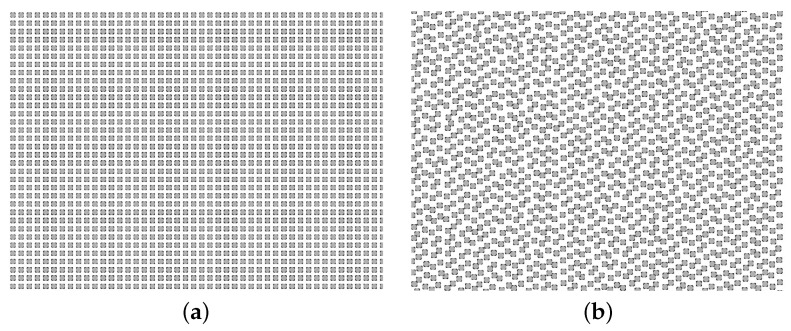
An example of a point cloud generated directly from the rasterization grid (**a**) and when using the sampling pattern of the prototype LiDAR (**b**). The grid-like pattern in (**a**) is the result of the equidistant pixels of the rendered output image. By contrast, LiDARs exhibit their own unique sampling pattern, which can be mimicked by associating uv-coordinates to each laser pulse. The uv-coordinates can be extracted by directing the real-world LiDAR perpendicular towards a flat surface, resulting in the patterns shown in (**b**).

**Figure 4 sensors-23-08130-f004:**
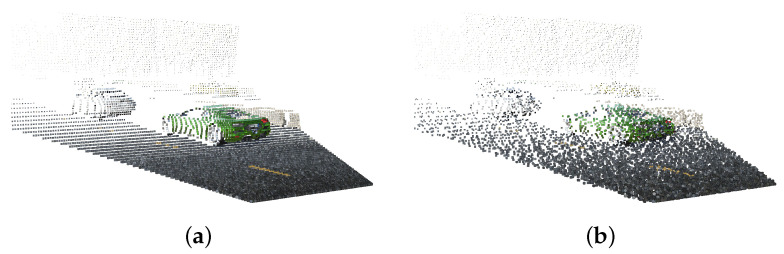
An example of a clean (**a**) and noisy (**b**) point cloud generated by the digital twin of our LiDAR. The clean point cloud directly results from reversing the rasterization pipeline of Figure 2. In reality, points captured by real-world LiDARs exhibit spatial noise in the form of displacements along their respective laser pulses, as shown in (**b**). This is the result of imprecise measurements and/or ambient light interaction or due to the reflectance properties of the hit surface.

**Figure 5 sensors-23-08130-f005:**
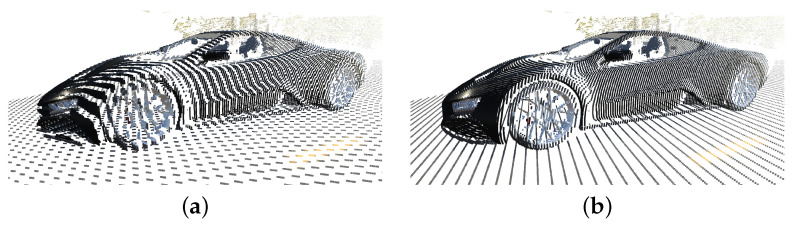
This figure demonstrates the impact of the bit-depth of the textures used for point cloud generation. Limiting the bit-depth in any phase of the rendering pipeline will result in aliasing artefacts (**a**). Employing 32 bits for the z-buffer and output textures produces clean point clouds (**b**).

**Figure 6 sensors-23-08130-f006:**
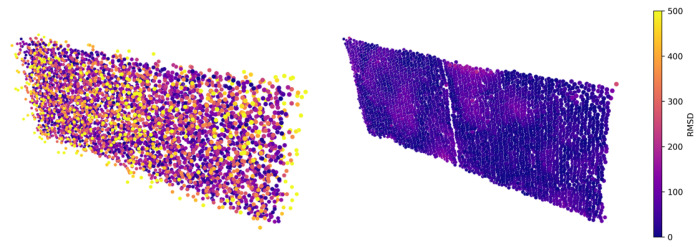
Visual results for our first experiment. The point clouds are captured by our real-world LiDAR for a calibration setup. Results before and after denoising are shown on the left and right, respectively. After denoising, the variance is reduced from 150.6 mm to 65.9 mm. Denoising was performed using a deep neural network trained with only synthetic data generated by our digital twin.

**Figure 7 sensors-23-08130-f007:**
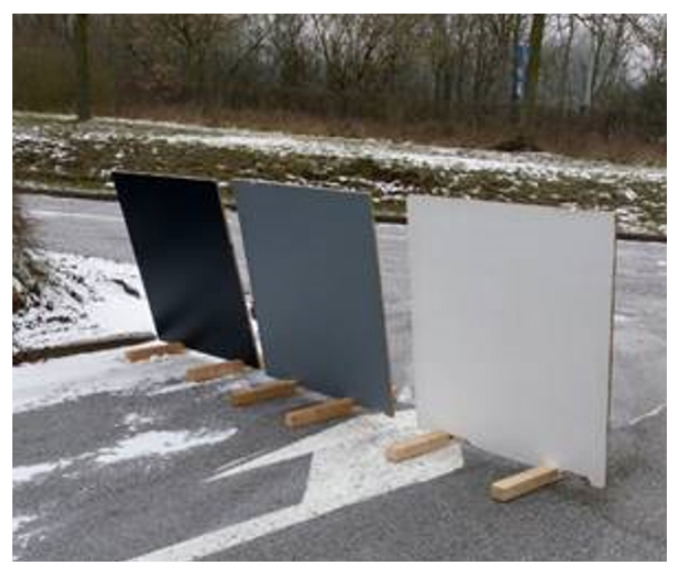
This figure shows the controlled environment for a second denoising experiment. The setup consists of 3 boards of different colours with known reflective properties and positioned at a distance of 100 m from the real-world LiDAR. Point clouds are captured and denoised using a neural network trained on synthetic data generated by our digital twin. Denoising results are shown in Table 2.

**Figure 8 sensors-23-08130-f008:**
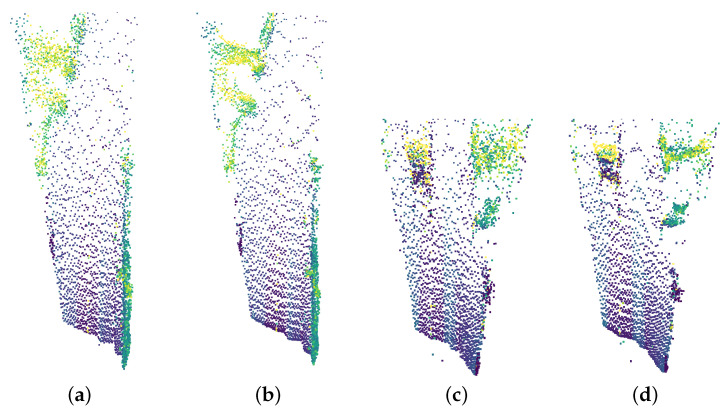
Bird’s-eye view of two frames captured by the real-world LiDAR. The colours indicate the intensity of the reflected laser beams. The noisy point clouds are shown in (**a**,**c**). The corresponding denoised frames are given in (**b**,**d**), respectively. Notice the more refined edges after denoising in (**b**), which reveals the open space in the top left corner. For the point cloud in (**d**), one can notice the vehicle at the top left becoming more defined. This is also true for the structures at the top right.

**Figure 9 sensors-23-08130-f009:**
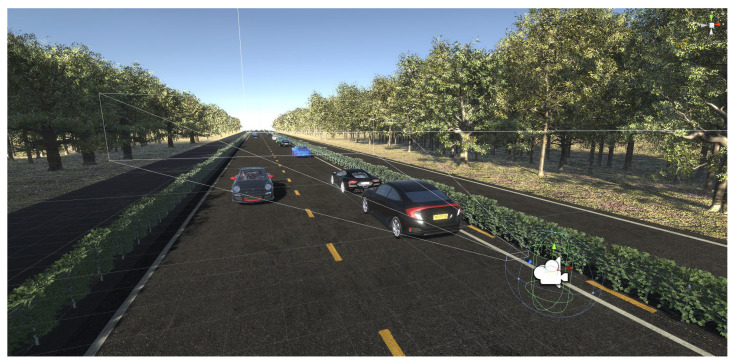
The proposed digital twin of our LiDAR is implemented in the Unity engine [71]. The figure demonstrates the scene used for data generation to train a neural network for semantic segmentation purposes. The spatial position of the camera coincides with that of the virtual LiDAR. The scene is modelled to resemble the environment of a real driving scenario captured using a real-world LiDAR, which is utilized for validating and testing the AI model.

**Figure 10 sensors-23-08130-f010:**
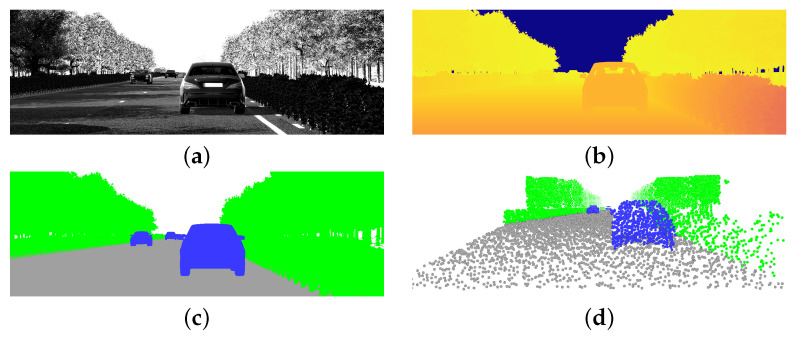
Data layers for point cloud generation using our digital twin implemented in Unity [71]. The luminance is visualised in (**a**). The corresponding depth and segmentation maps, shown in (**b**,**c**), are used to construct the auto-annotated point cloud of (**d**). Specifically, points are constructed by reversing the rasterization pipeline according to Equation (Equation 2) and subsequently mimicking the real-world LiDAR’s laser pattern by sampling the output textures (**b**,**c**) using the LiDAR’s associated uv-coordinates. Annotations are assigned on a nearest-neighbour-basis using the segmentation map shown in (**c**), resulting in a perfectly annotated point cloud according to the specifications of the real-world LiDAR.

**Figure 11 sensors-23-08130-f011:**
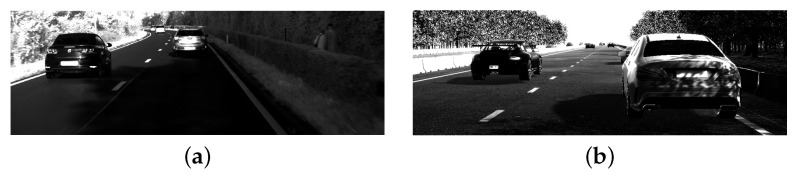
Comparison between images captured by the CMOS sensor of our real-world LiDAR (**a**) and its digital twin (**b**). Though the synthetic images are not used during training, they clearly demonstrate the resemblance of the output of the proposed digital twin and the real-world device. Resolution, field-of-view, sensor sensitivity, LiDAR position, and orientation are all accounted for during simulation, which is mandatory for successfully training neural networks operating on real data.

**Figure 12 sensors-23-08130-f012:**
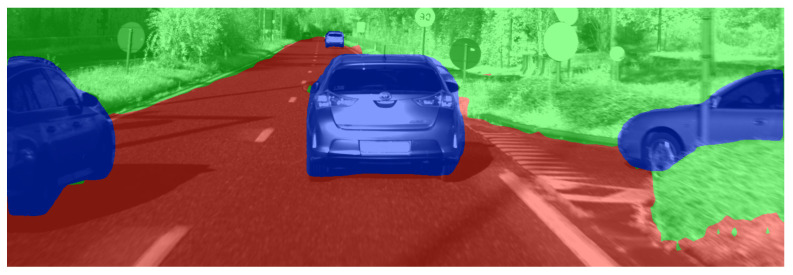
A representative real-world sample of the ground truth for the CMOS image captured by our LiDAR. The annotations are obtained using a retrained Xception 65 model [72] trained on the Cityscape dataset [14]. Note that annotations are not manually corrected, as this proved impractical. Annotated point clouds are obtained by transferring the labels from the image to the associated point clouds, which, in turn, serve as ground truth for model testing.

**Figure 13 sensors-23-08130-f013:**
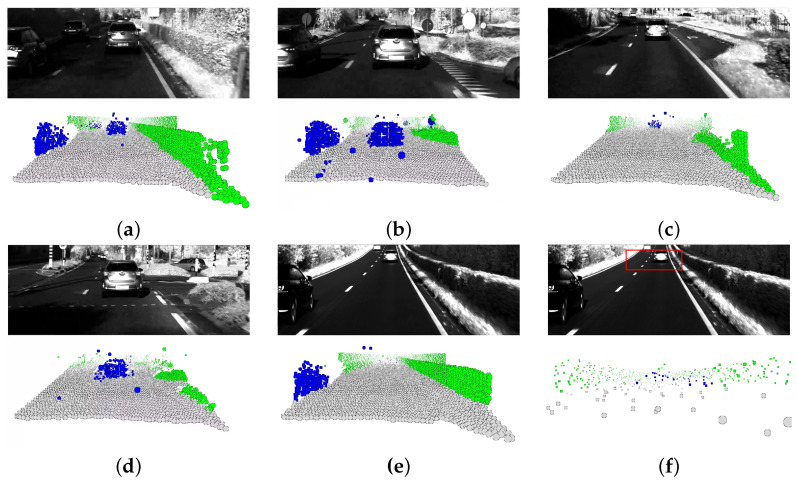
Visual segmentation results for data captured by our real-world LiDAR. The colours blue, gray, and green represent cars, the street, and flora, respectively. The employed model was obtained using the synthetic dataset enriched with 1200 real-world samples. The figures clearly demonstrate that the trained neural network performs well in segmenting point clouds obtained from the real-world LiDAR system. Notably, the network is capable of simultaneously detecting multiple cars, as evidenced in (**a**,**b**), and it accurately defines the boundaries of the street, even when it is not simply straight, as exemplified in (**c**,**d**). An interesting observation is that the network is able to distinguish between the street and cars with a high degree of accuracy using only a small subset of selected points, as demonstrated in (**f**), which is a zoomed-in section of the point cloud shown in (**e**) and corresponds with the area inside the red rectangle of (**f**).

**Figure 14 sensors-23-08130-f014:**
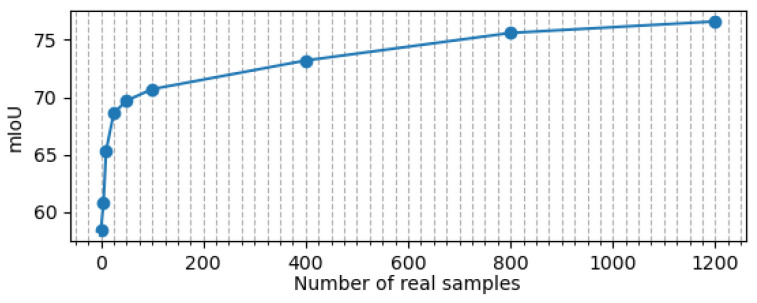
This figure illustrates the impact of the quantity of partially annotated real-world samples on the mean Intersection-over-Union (mIoU). The dots displayed in the graph correspond to different sample sizes, including [0, 5, 10, 25, 50, 100, 400, 800, 1200] samples. It is noteworthy that there is a substantial and rapid increase in the mIoU when relatively small quantities of real samples are introduced. For instance, the utilization of just 25 such samples results in a significant boost in mIoU, elevating it from 58.35% to 68.62%. However, as the number of real samples continues to increase, the rate of improvement gradually diminishes. When the dataset is enriched with 1200 real samples, an mIoU of 76.54% is achieved.

**Table 1 sensors-23-08130-t001:** Data generation speed comparison with the state-of-the-art for different number of simulated laser pulses. The numbers are expressed in milliseconds. Note that the proposed method reduces data generation time by a factor of 300 compared to Blainder [18], which is the state-of-the-art in ToF sensor simulation. For the state-of-the-art driving simulator Carla [19], execution times are reduced by a factor of 100. A custom implementation revealed that the proposed method still vastly outperforms a lightweight ray-casting-based ToF simulator relying on NVIDIA OptiX [69] for GPU acceleration.

	Simulated Laser Pulses				
**Simulator**	**1K**	**10K**	**100K**	**1M**	**10M**
Blainder [18]	45.20	79.76	566.73	5381.91	53,750.80
Carla [19]	4.24	5.78	109.75	1386.94	11,653.42
NVIDIA OptiX [69]	18.35	19.02	26.72	74.76	559.74
Proposed	0.61	0.81	2.27	10.87	113.41

**Table 2 sensors-23-08130-t002:** This table shows the standard deviation of the measured distance of the test setup shown in Figure 7. Results are provided in meters. Point clouds were captured by our real-world LiDAR from a distance of 100 m and denoised using the deep neural network trained solely on synthetic data. The experiment reveals that the neural network is successfully trained and reduces noise for all 3 coloured boards by approximately 20%.

Target	Before Denoising	After Denoising	Noise Reduction
Dark	6.54	5.24	19.88%
Gray	4.71	3.61	23.35%
Light	2.46	1.96	20.33%

**Table 3 sensors-23-08130-t003:** Quantitative semantic segmentation results for a driving scenario captured using our real-world LiDAR. We note that only cars are annotated for the real data. The model trained solely on such data is therefore unable to predict other classes. The table demonstrates that the network trained solely on synthetic point clouds achieves satisfactory results in semantic segmentation. Furthermore, this model outperforms the network trained on 800 real-world samples, specifically for the “car” class, in terms of accuracy. The best results are obtained when incorporating both synthetic and a small amount of real data using the proposed loss function defined in Equation (Equation 5).

Training Data		Results								
**Synth**	**Real**	**Car**		**Street**		**Flora**		**Average**		
		Acc	IoU	Acc	IoU	Acc	IoU	oAcc	mAcc	mIoU
10,000	0	77.38	32.72	72.48	67.83	**89.74**	74.71	79.90	78.86	58.35
0	800	74.02	50.81	-	-	-	-	-	-	-
10,000	800	**79.72**	**62.88**	**90.73**	**83.29**	89.18	**80.55**	**89.36**	**86.51**	**75.64**

## Data Availability

Not applicable.

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
