# Peer review of "GPU Rasterization-Based 3D LiDAR Simulation for Deep Learning"

_sensors, 2023, doi:10.3390/s23198130_

Round 1
Reviewer 1 Report
Summary:
The manuscript introduces a GPU-accelerated simulator designed for generating high-quality, labeled data for Time-of-Flight sensors, with a particular emphasis on LiDAR. The strength of this work lies in the use of the 3D graphics pipeline of the GPU to optimize data generation time, the development of generic algorithms that can simulate any ToF sensor, and the introduction of a novel loss function that bridges the gap between synthetic and real data. This work addresses the critical issue of data acquisition for deep-learning applications, particularly in the context of autonomous vehicles.
General Concept Comments:
The article effectively tackles the problem of acquiring and annotating LiDAR-specific data. The authors provide a well-reasoned argument for the advantages of using simulated data over real-world data. This is an area of significance given the increasing role of deep learning in autonomous vehicle systems. While the introduction provides a comprehensive background, some of the cited references could be reviewed for relevance. It might benefit from referencing more recent publications, ensuring a balanced representation of the field without excessive self-citations. The presented algorithms, while promising, would benefit from a more detailed explanation in the methods section. While the methodology's broad strokes are clear, the specifics that would allow for reproducibility and validation by peers might need further elaboration.
Additional Points to Consider:
For the paper's broader audience, a section that provides a basic understanding of Time-of-Flight sensors and LiDAR might be beneficial. The data's reproducibility based on the described methods is a crucial aspect. If not present in the subsequent sections, providing a more granular breakdown of the methods will be helpful. Ethics and data availability statements should be revisited to ensure their adequacy, especially given the increasing emphasis on data transparency in AI research.
In conclusion, the article addresses a significant challenge in the realm of deep-learning applications, providing a novel solution that has the potential for broad impacts, especially in the autonomous vehicles domain. This paper will make a valuable contribution to the field.
Author Response
We thank you very much for your thorough evaluation of our manuscript. Please find attached our responses to your comments.
Sincerely,
Leon

Reviewer 2 Report
This manuscript sensors-2586835 presents a GPU-accelerated simulator that enables generating high-quality perfectly labelled data for any Time-of-Flight sensor, including LiDAR. Our approach optimally exploits the 3D graphics pipeline of the GPU, significantly decreasing data generation time while preserving compatibility with all real-time rendering engines. The presented algorithms are generic and allow to perfectly mimic the unique sampling pattern of any such sensor. To validate our simulator, two neural networks are trained for denoising and semantic segmentation. To bridge the gap between reality and simulation, a novel loss function is introduced that requires only a small set of partially annotated real data. It enables the learning of classes for which no labels are provided in the real data, hence, dramatically reducing annotation efforts. With this work, we hope to provide means for alleviating the data acquisition problem which is pertinent to deep-learning applications. It was a pleasure reviewing this work and I can recommend it for publication in Sensors after a major revision. I respectfully refer the authors to my comments below.
1. The English needs to be revised throughout. The authors should pay attention to the spelling and grammar throughout this work. I would only respectfully recommend that the authors perform this revision or seek the help of someone who can aid the authors.
2. (Section 1 Introduction) The reviewer hopes the introduction section in this paper can introduce more studies in recent years. The reviewer suggests authors don't list a lot of related tasks directly. It is better to select some representative and related literature or models to introduce with certain logic. For example, the latter model is an improvement on one aspect of the former model.
3. Experimental pictures or tables should be described and the results should be analyzed in the picture description so that readers can clearly know the meaning without looking at the body.
4. (Section I, Introduction) The reviewer suggest to revise the original statement as “The field of AI has witnessed rapid development in recent years [*-***]” (High-resolution facial expression image restoration via adaptive total variation regularization for classroom learning environment," Infrared Physics & Technology, 2023.)”
5. (Tables 1 and Table 2) All the values in this table should be with same data accuracy. The number of data after the decimal point are the same. Please check other Tables and sections.
6. (Section I, Introduction) The reviewer suggest to revise the original statement as “This is particularly important for data-driven applications such as deep-learning [*] …” ([*]DOI: 10.1109/TMM.2023.3238548)
7. The authors are suggested to add some experiments with the methods proposed in other literatures, then compare these results with yours, rather than just comparing the methods proposed by yourself on different models.
8. Discuss the pros and cons of the proposed models.
My overall impression of this manuscript is that it is in general well-organized. The work seems interesting and the technical contributions are solid. I would like to check the revised manuscript again.
The English needs to be revised throughout. The authors should pay attention to the spelling and grammar throughout this work. I would only respectfully recommend that the authors perform this revision or seek the help of someone who can aid the authors.
Author Response

(The authors gave the same response as above.)

Reviewer 3 Report
1. The writing should be improved, e.g., line 83 “Most recent works include [17–25].”
2. How to get (3)? From other studies or by the author?
3. It is very common that such sensor models as camera and lidar run in GPU. Is there any new in section 3.4?
4. What is the unit of Tab. 1 and 2?
5. There’s already many commercial software (Open source or not), such as Prescan, VTD, Carla and etc. It is suggested to compare with these ones. These sensor models are also calculated by GPU and what’s the contribution of this study?
6. The effectiveness of the proposed model should be validated by comparing with the point cloud from a real lidar, not just by different simulation data.
Some should be improved and please see the comments for author.
Author Response

(The authors gave the same response as above.)

Round 2
Reviewer 3 Report
The author has addressed all my concerns.